# QUANTIZATION-AWARE POLICY DISTILLATION (QPD)

## ABSTRACT

Recent advancements have made Deep Reinforcement Learning (DRL) exceedingly more powerful, but the produced models remain very computationally complex and therefore difficult to deploy on edge devices. Compression methods such as quantization and distillation can be used to increase the applicability of DRL models on these low-power edge devices by decreasing the necessary precision and number of operations respectively. Training in low-precision is notoriously less stable however, which is amplified by the decrease in representational power when limiting the number of trainable parameters. We propose Quantization-aware Policy Distillation (QPD), which overcomes this instability by providing a smoother transition from high to low-precision network parameters. A new distillation loss specifically designed for the compression of actor-critic networks is also defined, resulting in a higher accuracy after compression. Our experiments show that these combined methods can effectively compress a policy network down to 0.5% of its original size, without any loss in performance.

## 1 INTRODUCTION

Deep Reinforcement Learning (DRL) recently achieved super-human performance on Atari games (Mnih et al., 2015), Go (Schrittwieser et al., 2020) and Starcraft (Vinyals et al., 2019). But at the same time, their policy networks have become significantly bigger. Running inference for such a network, such as ResNet-200(He et al., 2016), can take half a second on a GPU and a large amount of memory. Applying such a network locally, on a low-power Locobot (loc, 2019) for example, becomes completely impractical however due to computation and power constraints. Model compression can counter this by reducing the size of Deep Neural Networks (DNN) without decreasing performance.

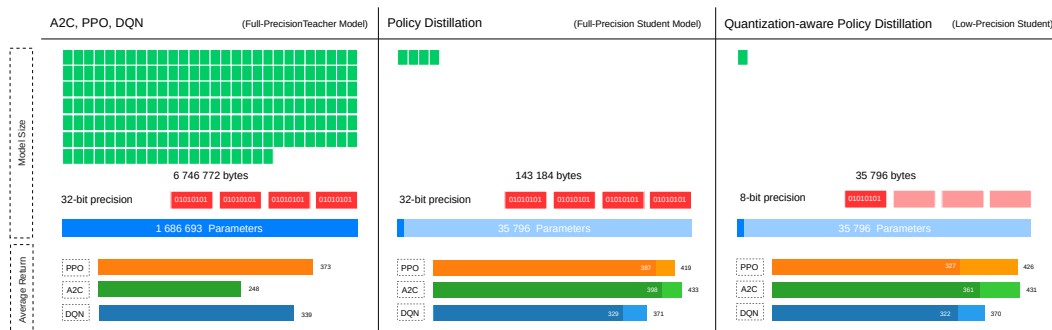

Figure 1: The size and performance differences between the original teachers, our policy distillation methods and QPD technique. The average returns on the Atari Breakout environment are shown for the best and worst performing student size.

Three such methods currently show the most potential: pruning, quantization and distillation. We introduce the Quantization-aware Policy Distillation (QPD) algorithm based on the latter two techniques, with the concept and results shown in figure 1. In quantization, the precision of the DNN parameters is reduced, requiring less memory and enabling inference on simpler embedded hardware (Stanton et al., 2021). Distillation can reduce the number of DNN parameters and therefore computations, by transferring knowledge of a larger teacher network to a student network with fewer

parameters. Through QPD, we create a model with (4x) smaller and (up to 47x) fewer parameters, while still maintaining and even exceeding the performance of the teacher model. Compressing DRL models enables low-power devices to perform inference using these models on the edge, increasing their applicability, reducing cost, enabling real-time execution and providing more privacy.

Our main contributions are threefold. (1) We propose a new distillation loss for actor-critic based teacher networks, with an auxiliary component for distilling state-value predictions, which improved the internal representations and average return obtained by the students, without increasing overhead for the final models. We also smoothen the teacher outputs to transfer more secondary knowledge, due to the stochastic policy of policy-gradient teachers. (2) We outline a novel method (QPD), for quantizing DRL networks using this loss, that provides a smoother transition from high to low-precision weights, which is able to overcome the unstable optimization that is encountered when training directly in low-precision. (3) We demonstrate how well different DRL teacher algorithms are suited for distillation under varying constrained conditions, including limited parameter count, precision and both combined. These results indicate that the choice of teacher has a larger impact than simply how well they perform themselves, and that the best suited teacher type depends on what constraints are in place.

## 2 BACKGROUND

### 2.1 DISTILLATION

In supervised Knowledge Distillation (KD), a DNN is compressed by training a small student network to emulate the larger teacher's outputs, which contains valuable secondary 'dark' knowledge Hinton et al. (2015) expressed in all the outputs, instead of only learning directly from the single 'correct' label for each sample. In DRL there is no set of labelled data however, so Rusu et al. (2016) proposed to record the observations and network outputs during on-policy interactions with the environment in a replay memory ($D$). This replay memory is periodically refreshed to widen the distribution of states encountered by the student. The student is then trained using the Kullback-Leibler divergence (KL) between the teacher ($q^T$) and student ($q^S$) outputs, with $\theta_S$ the trainable student parameters and $\tau$ a temperature used to sharpen or smoothen the teacher outputs:

$$L_{KL}(D, \theta_S) = \sum_{i=1}^{|D|} \text{softmax}(\frac{q_i^T}{\tau}) \ln(\frac{\text{softmax}(\frac{q_i^T}{\tau})}{\text{softmax}(q_i^S)}) \tag{1}$$

Instead of training a smaller network directly, the student only needs to learn how to follow the final teacher policy, while the teacher still contains redundant exploration knowledge about non-optimal trajectories. We argue that this knowledge is necessary to find the optimal policy, but not to follow it, so it can be omitted from the student. Using overcomplete DRL models also helps with alleviating optimization issues, such as getting stuck in local minima, which occur less when learning to emulate an existing network in distillation (Rusu et al., 2016).

### 2.2 QUANTIZATION

In quantization, models are compressed by training reduced precision parameters (e.g. 8-bit instead of 32-bit). Optimizing low precision parameters directly is unstable however, so a transformation from high to low-precision representations is often used instead (Mishra & Marr, 2018). In Post-Training Quantization (PTQ), a full-precision model is transformed into low-precision after training, while approximately preserving its behaviour (Gholami et al., 2021). This introduces inaccuracies that can accumulate when propagating forward through the network, so Quantization-Aware Training (QAT) is often preferred instead to account for these, where the quantization transformation is part of the architecture while training. Quantization functions can be linear, where the relative distance between values in the original representation is generally maintained after quantization, or non-linear, where this is not the case. This property is required for (de-)quantizing the network inputs and outputs, and in combination with PTQ for network parameters. When using QAT, non-linear quantization can be preferred however, because there certain regions in the original representation are represented more accurately than others, more closely matching the distribution of values that need to be quantized (Gholami et al., 2021). We therefore apply both types in section 4.2, and explain the functioning and reasoning behind the used methods further.

## 3 RELATED WORK

This work relates to three main research areas: policy distillation, quantization through distillation and quantization of DRL policies. The following sections highlight not only how our approach differs from previous work in these areas individually, but we also successfully combine them.

**Policy Distillation** The concept of KD was first applied for compression of DRL policies by Rusu et al. (2016). They demonstrated that through policy distillation, students could be trained with only 7% the size of their DQN teacher, but still obtain higher scores for 10 Atari games. This shows that distillation to a reduced capacity model acts as a form of regularization. Green et al. (2019) extended this to a PPO teacher by removing the temperature $\tau$ from the distillation loss, as the policy-gradient teacher outputs are less uniformly distributed than for a value-based method, and therefore did not need to be sharpened. They also fine-tuned their students using regular PPO after training, which further increased performance on some environments. Our work extends on this by investigating the effects of the teacher algorithm choice more closely, including for an additional A2C teacher. We also study how the temperature parameter influences the actual teacher output distributions, leading to different conclusion of smoothening the distribution instead. Finally, we look at how to use the knowledge from the critic head of an actor-critic teacher to further enhance the internal policy representation, whereas Green et al. (2019) only considered the actor head for distillation.

**Quantization through Knowledge Distillation** Applying KD in low-precision has only been done in supervised learning, so our work differs by applying it for DRL instead. Mishra & Marr (2018) proposed to initialize low-precision student weights with the full-precision weights of a teacher with the same size and architecture, and fine-tuning them using KD. We initialize our quantized weights similarly, except that we apply PTQ on a full-precision distilled student instead of on the teacher, as our students are smaller than their teachers. Kim et al. (2019) made the transfer of knowledge smoother by first training the teacher to behave more similarly to the student using a combined loss with the true labels and each other's distributions. They argue that these changes are necessary due to the regularization effect of distillation further diminishing the already poorer representational power of a quantized model. Their methods do not translate well to a DRL setting however, where there is no set of true labels. QPD solves the same issue by using a similar concept of smoothening the knowledge transfer, but without relying on labelled data.

**Policy Quantization** Preliminary work on the quantization of DRL policies was done by Krishnan et al. (2019) (QuaRL). They successfully applied PTQ and QAT using standard uniform affine quantization on policies trained using various DRL algorithms (A2C, DDPG, DQN, D4PG, PPO), resulting in a relative error between 2% and 5% for 8-bit PTQ and no performance loss down to 6-bit QAT. Björck et al. (2021) were instead able to train a DRL policy directly in a native 16-bit floating point format using SAC, instead of using a quantization function. 6 Improvements to the numerical stability of SAC and the Adam optimizer were necessary to make training as stable as in 32-bit floating point. Our work differs by using policy distillation for training 32-bit parameters to initialize the 8-bit ones and to continue training after quantization using the same state distribution, in addition to enabling even further compression by reducing the parameter count. We also use a non-linear function in both PTQ and QAT for quantizing the trainable parameters.

## 4 METHODOLOGY

### 4.1 DISTILLATION OF ACTOR-CRITIC NETWORKS

Rusu et al. (2016) initially proposed policy distillation for DQN teachers, which model a state-action value (Q) function. Actor-critic algorithms, such as PPO (Schulman et al., 2017) and A2C (Mnih et al., 2016), operate by modelling the policy ($\pi$) directly instead. There, the network output consists of two heads: an actor and a critic. The actor models the policy explicitly using a probability distribution over the possible actions. The critic on the other hand is a state-value (V) function that provides a direction in which to update the actor during training. The stochasticity of the actor-critic policy needs to be maintained when transferred to the student, while for a value-based teacher the student only needs to learn to choose the same deterministic actions. Due to this difference between learning a value function or an explicit policy, we show that some teachers are more compatible with the distillation process and have representations that are more easily transferable to a lower-precision student, but that this also requires changes to how the teacher outputs are distilled into the student.

### 4.1.1 ACTOR SMOOTHING

Considering the difference between how learning an explicit policy and a value function is expressed in the network outputs, we first explore the characteristics of the Q-value predictions in a DQN and compare this to the action-probability logits of an A2C model. The Q-values predicted by a DQN teacher are often very similar to the other action-values for the same state, since the value of being in a certain state in the environment does not usually change significantly when choosing a suboptimal action once, partly due to the ability for the agent to correct a mistake and still reach the goal. Small inaccuracies in the predicted action-values can therefore have a large impact on the effectiveness of the trained student network, although the loss for these actions is relatively small. Such DQN outputs can be seen in figure 2a, while in figure 2b the softmax in equation 1 is applied with $\tau = 1$. Rusu et al. (2016) therefore choose a temperature value $\tau < 1$ in equation 1, the effects of which can be seen in figure 2c. They sharpen the DQN outputs to make the differences between the possible actions larger, thereby increasing the likelihood of the student selecting the correct one.

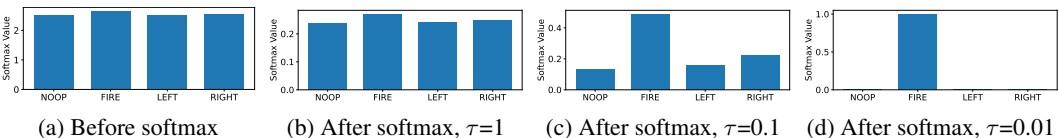

(a) Before softmax     (b) After softmax, $\tau$=1     (c) After softmax, $\tau$=0.1     (d) After softmax, $\tau$=0.01

Figure 2: Encountered DQN teacher outputs, before and after softmax and with different $\tau$ values.

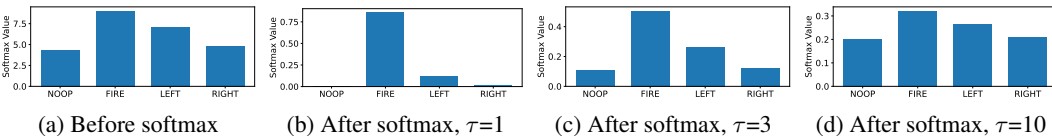

(a) Before softmax     (b) After softmax, $\tau$=1     (c) After softmax, $\tau$=3     (d) After softmax, $\tau$=10

Figure 3: Encountered A2C teacher outputs, before and after softmax and with different $\tau$ values.

In contrast, the outputs of policy-gradient teachers are already more peaked, as can be seen in figure 3b. But these teacher types suffer from the opposite problem during distillation. The loss of the action with the highest probability dominates over the other actions, resulting in little secondary knowledge being transferred. Learning this secondary knowledge is critical for generalization, as it provides insight for recognizing similar situations and viable alternative trajectories. If the action-probabilities are similar, the policy determined it is beneficial to choose these actions similarly often, so transferring knowledge on all viable actions to the student is essential. Rusu et al. (2016) empirically chose $\tau = 0.01$ for DQN teachers, resulting in basically all secondary information being lost, as seen in figure 2d. For policy gradient based teachers, we suggest using $\tau \in [1,5]$ instead, so more secondary knowledge is transferred, but alternative actions are not overvalued significantly. We therefore use $\tau = 3$ in our experiments discussed in section 6.

### 4.1.2 AUXILIARY CRITIC LOSS

In addition to distilling the actual policy, we explore learning an auxiliary loss for the critic, that contains additional dark knowledge. This critic is often learned through a dual-head architecture, because there is overlapping logic required to compute both heads. Distilling the state values in the student will likely have a similar benefit to the internal representation of the policy. Learning from more data should also increase sample efficiency and increase teacher similarity. We therefore propose a new distillation loss for dual-headed students:

$$L_{\text{Distillation}}(D, \theta_S) = \left(\lambda \frac{L_{\text{Actor}}}{v(L_{\text{Actor}})} + (1 - \lambda) \frac{L_{\text{Critic}}}{v(L_{\text{Critic}})}\right) * (v(L_{\text{Actor}}) + v(L_{\text{Critic}})) \tag{2}$$

$$L_{\text{Actor}} = \sum_{i=1}^{|D|} \text{softmax}(\frac{a_i^T}{\tau}) \ln(\frac{\text{softmax}(\frac{a_i^T}{\tau})}{\text{softmax}(a_i^S)}) \quad \text{and} \quad L_{\text{Critic}} = \sum_{i=1}^{|D|} L_{\text{Huber}}(c_i^T - c_i^S) \tag{3}$$

Here is $a_i$ the outputs of the actor head, $c_i$ those of the critic, $\lambda$ a parameter for tuning the relative importance of each head and $v$ a function that returns the value of a scalar, while decoupling it

from the backward propagation as if it were a constant. The regular distillation loss based on the Kullback-Leibler (KL) divergence is used for the actor outputs, while the critic values are distilled using a Huber loss. The critic head returns a single value for each state as opposed to a distribution over actions for the actor, so the KL divergence that is typically used in distillation is not well-suited for transferring this type of value. A composition of different loss functions is therefore needed, as is done in equation 2. By normalizing both sub-losses before combining them, we ensure the loss from one of the heads does not heavily dominate over the other in the weight updates at any point during training. Due to the differences in the magnitude and type of loss functions, they decrease at different rates, meaning that a simple linear combination would not yield the same desired effect for each update step. But through normalization, the loss from each head is always equally represented in the combined loss (when $\lambda = 0.5$). The multiplier in equation 2 is introduced to ensure that our loss theoretically decreases monotonically for each update step, a property that is required for some optimizers. Using $\lambda$, we can adjust how much each head influences the internal representation of our student model, with 1 corresponding to only learning the action-probablities (i.e. regular distillation) and 0.5 corresponding to learning from both heads equally. Prioritizing the critic loss too heavily could also be detrimental, as this only serves to improve the internal representation, while only the actor head is essential to follow the actual policy. This also means that, after training, the critic head can be completely pruned from the architecture, resulting in no overhead in terms of model size compared to vanilla policy distillation.

## 4.2 QUANTIZATION-AWARE POLICY DISTILLATION (QPD)

The combination of quantization and policy distillation could allow a network to be compressed significantly more than simply applying both techniques separately. Both techniques show diminishing returns for very small networks, either because of excessive quantization noise (Krishnan et al., 2019) or limited representational power respectively. But combining these methods before reaching their respective limit could yield considerably more compression while still having viable performance. Quantization and policy distillation also have distinct non-overlapping benefits, which can now both be taken advantage of (see section 2). Optimization in DRL is generally less stable than in supervised learning, which is only amplified when working in low-precision (Björck et al., 2021). QPD alleviates this, partly by being more similar to supervised than reinforcement learning in terms of loss optimization. Increased precision to accurately represent the slight advantage of certain trajectories can also be beneficial to converging on a more optimal policy, but this is less necessary when emulating an existing policy. Simply training a low-precision student based on a full-precision teacher network through policy distillation fails to converge to any sensible policy however (see appendix A.3 for this experiment), due to the reduced representational power of the quantized model failing to accommodate for the regularization effect of policy distillation, which was also observed by Kim et al. (2019) in a supervised DL setting. To address these issues, we propose the QPD algorithm 1, which applies a smoother transition from the full-precision parameters of the teacher to the low-precision student parameters.

### 4.2.1 ALGORITHM OVERVIEW

This smoother transition is done by first training a set of full-precision student parameters ($\theta_f$) using our policy distillation methods discussed in section 4.1 (phase 1), which serves as a good initialization for the low-precision parameters ($\theta_q$) by applying PTQ (phase 2). The loss when training with randomly initialized low-precision parameters is very steep, as large adjustments are necessary, which is made infeasible by the optimization instability due to quantization. Quantizing the full-precision parameters and using them directly is also not possible in combination with the non-linear quantization method, but they can be used as part of the quantization function when introduced in the computation graph of the network (phase 3). These quantized outputs are already much closer to the final low-precision parameter values compared to using randomly initialized values. The state distribution encountered before and after the second phase remains identical, since the training data in policy distillation is independent of the student behaviour, making converging back to the already discovered solution easier. After this transformation, the low-precision parameters mainly need to be adjusted to account for the non-linear distribution shift and to reduce the effects of noise caused by the loss of precision, which is significantly more stable than training from scratch. We continue to use and adapt the full-precision parameters $\theta_f$ to optimize the low-precision parameters $\theta_q$ while training (QAT). Forward passes are performed based on $\theta_q$ to compute the gradient of the distillation loss $\nabla L$

1 in full-precision. We then update $\theta_f$ based on this gradient through the use of a *straight-through estimator (STE)* to circumvent the non-differential property of the quantization function, which consists of simply using the gradient for $\theta_q$ to optimize $\theta_f$ as-is. After $\theta_f$ is updated, we recompute the values for $\theta_q$, such that the gradient updates are also propagated to the low-precision parameters for the next update step. This procedure is then repeated until the student using $\theta_q$ stops improving on the DRL environment.

---

**Algorithm 1:** Quantization-aware Policy Distillation

---

Randomly initialize $\theta_f$ of $M_s$ and fill replay buffer $D$ using interactions between $M_t$ and *env*;

```
/* Phase 1:  Train a student with full-precision parameters   */
```
**while** $M_s$ *has not converged* **do**
    **for** $i \leftarrow 0$ **to** *update steps* **do**
        Sample $D_i \subset D$ ;
        $\theta_f \leftarrow$ Update $\theta_f$ using equation 2 and $D_i$;
    **end**
    $D \leftarrow$ Update oldest($D_o \subset D$ using new interactions between $M_t$ and *env*;
**end**

```
/* Phase 2:  Quantize the distilled student network (PTQ)     */
```
$\theta_q \leftarrow$ Quantize $\theta_f$ using equation 5;

```
/* Phase 3:  Continue training with quantized weights (QAT)   */
```
**while** $M_s$ *has not converged* **do**
    **for** $i \leftarrow 0$ **to** *update steps* **do**
        Sample $D_i \subset D$ and quantize it using equation 4;
        Perform forward pass using equation 2, memory $D_i$ and parameters $\theta_q$;
        Compute full-precision gradient $\nabla L$ based on low-precision activations;
        $\theta_f \leftarrow$ Update($\theta_f, \nabla L$) ;
        $\theta_q \leftarrow$ Quantize $\theta_f$ using equation 5 (QAT);
    **end**
**end**

---

### 4.2.2 QUANTIZATION FUNCTIONS

Both linear and more complex non-linear quantization functions are used in QPD, each for a different value type. The inputs of the first network layer and the activations of the final layer need to maintain their approximate value after the transformation, for which we use standard uniform affine quantization. This is a linear function where the range of values in the original representation is rescaled to $[0, 2^k - 1]$ as follows (Zmora et al., 2019):

$$w_q = round\Big(\frac{(w_f - \min(w_f))(2^k - 1)}{\max(w_f) - \min(w_f)}\Big) \tag{4}$$

Here is $w_f$ is the original full-precision representation and $w_q$ the resulting value represented using only $k$ bits of precision. During the second and third phase, we apply a non-linear method proposed by Zhou et al. (2016) (DoReFa) to create a set of quantized parameters $\{w_q \in \theta_q\}$ based on the full-precision parameters $\{w_f \in \theta_f\}$ computed during the first phase:

$$w_q = 2\Big(\frac{round((2^k - 1)f(w_f))}{2^k - 1}\Big) - 1 \quad \text{with} \quad f(w) = \frac{\tanh(w)}{2\max_{w_i \in \theta_f}(|\tanh_{w_i}|)} + \frac{1}{2} \tag{5}$$

## 5 EXPERIMENTAL SETUP

With reproducibility in mind, we use pre-trained Stable Baselines3 (SB3) (Raffin et al., 2019) Breakout agents (DQN, A2C, PPO) for our teachers. A trade-off was made by focussing on this environment to provide a more in-depth investigation into the impact of different teachers and student sizes, instead of repeating the same experiment for more environments. These teachers obtain an average return across 200 episodes of 373, 248 and 339 respectively. This difference is already substantial, so the student results need to be compared both in absolute performance and relative to their respective teacher. In particular, the A2C checkpoint significantly underperforms, as it should be comparable to PPO (Schulman et al., 2017), but this published checkpoint already started to regress. Training the A2C teacher ourselves yielded a mean return of 342, but our distillation results did not change significantly when used instead (see the appendix A.1). We therefore decided to use the SB3 checkpoint for better reproducibility and to highlight this counter-intuitive result that clearly demonstrates the regularization effect of policy distillation. We consider 7 sizes for the student networks, as shown in table 3 in the appendix, ranging from exceeding the teacher size (XXL), to being almost fifty times smaller (XXS). All teachers have roughly the same size as the XL student, so the level of compression is independent of the used teacher model. For each (student size, teacher algorithm) combination, we collect the average return of 50 episodes for each of 600 training epochs. A training epoch is completed when the student has seen all 540,000 transitions stored in the replay memory ($D$). Transitions are sampled in a random order from D in sets of mini-batches. After each epoch, the oldest 10% of transitions in D are replaced through new teacher-environment interactions. Each experiment configuration is repeated using 5 independent runs and the collected average return for the best student of each run is again averaged.

## 6 RESULTS AND DISCUSSION

### 6.1 ACTOR CRITIC ARCHITECTURES

Figure 4 shows a clear benefit of distilling the critic head in addition to the actor. Students that learned a state-value function consistently outperformed those without our proposed loss. The exact choice of $\lambda$ does not significantly influence the results, so no extensive hyperparameter search is necessary, as long as the model is able to learn from it in some degree.

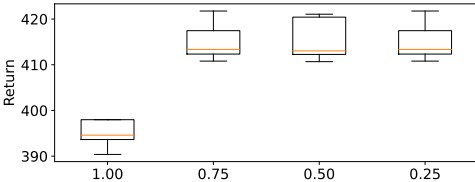

Figure 4: Distribution of returns for 5 runs with student size S, A2C teacher and 4 values for $\lambda$.

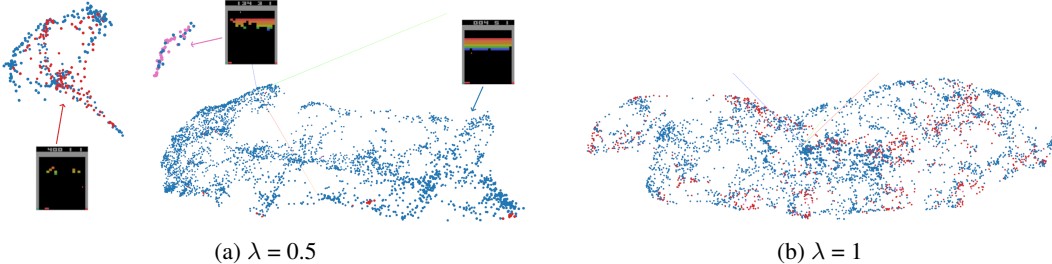

(a) $\lambda = 0.5$    (b) $\lambda = 1$

Figure 5: UMAP plots of the activations of the last shared layer between the actor and critic head. Activations with low, medium and high critic values were assigned red, blue and pink respectively.

We also investigated the impact of this auxiliary loss on the internal representation of our students by creating UMAP (McInnes et al., 2018) plots (figure 5) of the activations of the last shared network

layer between the actor and critic head. Three distinct clusters appear in the activation values when learning the state values ($\lambda = 0.5$), but not when using the regular distillation loss ($\lambda = 1$). These clusters clearly match with a particular range of critic values, as indicated by the colours. The red cluster mostly contains states with very few blocks remaining, so little expected reward is still to be received. States in the pink cluster correspond the ball reaching above the blocks for a continued period of time (the tunnel strategy), resulting in a large short-term return. We conclude that distilling the critic as an auxiliary loss noticeably impacts the internal representation, leading to a clear improvement on the chosen task, without increasing the final model size.

## 6.2 TEACHER ALGORITHMS & STUDENT SIZES

We next investigate how the teacher type and student parameter count affects distillation performance. Our actor-critic loss from section 4.1 using PPO and A2C teachers is compared to a DQN baseline with vanilla policy distillation to determine whether either an explicit policy or an action-value function respectively is more compatible with distillation. The effectiveness of our proposed distillation changes are first studied in isolation to later compare with QPD in section 6.3.

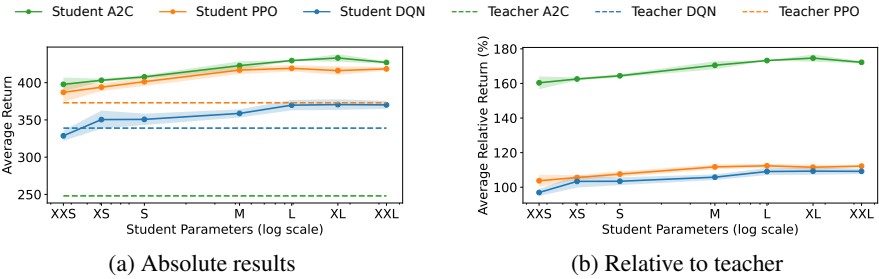

(a) Absolute results                    (b) Relative to teacher

Figure 6: Average return for student networks of different sizes and teacher models.

The PPO & A2C students clearly see the best absolute results, as seen in figure 6a, suggesting actor-critic methods were more suited for distillation than the value-based DQN teacher. We hypothesize that this is partly due to a smoother loss for learning a probability distribution than for action-values. Our new actor-critic loss also provides students additional state-values to learn from. Action-values from a DQN contain similar information, but this is lost when applying the softmax function and temperature scaling in equation 1. The DQN teacher was also already weaker to begin with, so less useful knowledge can be transferred, as seen in figure 6b, where the DQN students perform more similarly to the PPO ones when viewed relative to their teachers. The relative improvement of the PPO students is still measurably higher however. DQN distillation suffers more from the limited capacity as well, with the XXS size performing worse than its teacher. All students eventually outperform their teacher by between 9% and 90% on this environment, while being up to 47 times smaller. This is a clear example of the regularization effect of distillation, as was also observed by Rusu et al. (2016), especially for the A2C students. The A2C teacher could no longer follow its optimal policy due to regression, but the students were able to recover through this effect. This is in contrast to the DQN teacher, which never learned a value function for such higher scores, so it cannot be transferred to a student. All our PPO students significantly outperform even the largest student (similar to our M in size) in the actor-distillation results by Green et al. (2019), both compared to their absolute mean return of 248 and relative to their teacher with 277. We also conclude that our proposed distillation loss can outperform vanilla policy distillation, both in terms of absolute and relative improvements. The appendix A.2 shows how this compression effectively translates to a real-world inference speed-up.

## 6.3 QUANTIZATION-AWARE POLICY DISTILLATION (QPD)

Finally, we investigate the effectiveness of our QPD algorithm, now resulting in a low-precision student. The used teacher could be even more impactful here, due to the limited precision to represent either an action-probability or action-value distribution, as well as the difference in KL-divergence and Huber loss as part of our actor-critic distillation loss (equation 2). Varying the number of student parameters might also have a larger influence, as the representational power is now limited by both the

reduced parameter count and precision. Looking at figure 7, especially the policy-gradient teachers indeed suffer significantly from the reduced precision for the smallest networks. For both teachers, the average return drops by 9-13% going from S to XS and again 8% from XS to XXS size, compared to roughly 2% for the DQN teacher and our full-precision results 6a. However, starting from the S size (15x compression), performance mostly recovers to a level comparable to our 32-bit precision distillation, as seen in figure 7c. Krishnan et al. (2019) (QuaRL) report a score below 400 using 8-bit QAT for both PPO and A2C, which our students outperform starting at size S for A2C and M for PPO, without even considering that we reduce the parameter count in addition to the precision. We conclude that QPD is effective at training low-precision DRL models, without any clear performance detriment for networks with a sufficient parameter count to compensate for the reduced representational power of working in low-precision. The total network size in terms of bytes is still 28% lower for the S quantized model than the XXS full-precision model, while performing significantly better on this environment. The quantized model can additionally be deployed much more efficiently on embedded hardware. A trade-off is therefore to be made between parameter precision and count, depending on the available hardware optimizations.

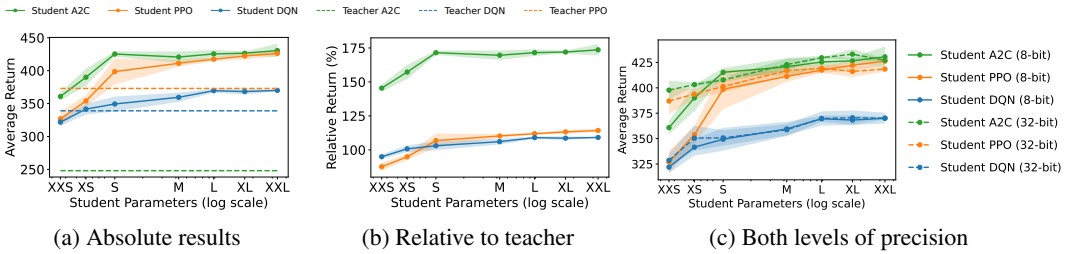

(a) Absolute results      (b) Relative to teacher      (c) Both levels of precision

Figure 7: Average return for students of varying sizes and teachers after applying QPD (algorithm 1).

# 7   CONCLUSIONS

A DRL policy can be compressed by either reducing the parameter count or precision. We developed the novel QPD method that employs both compression types, namely policy distillation and quantization. To achieve this, we first proposed a new distillation loss for actor-critic teachers, through which the student learns from the critic head in addition to the actor, yielding a 7% average return improvement without increasing the model size. We also discussed how the temperature in the distillation loss can be tweaked for policy-gradient teachers, such that more secondary knowledge is transferred to the student. These changes outperformed vanilla DQN policy distillation, suggesting that actor-critic teachers can be more effectively distilled. QPD uses these improvements to first train a set of full-precision parameters, which are then transformed to low-precision (PTQ) and continuously optimized for this transformation by including a quantization function as part of the computation graph (QAT). Results demonstrated this approach to be very effective, without any loss in performance compared to full-precision when using networks with a sufficient number of parameters. Using QPD, it is possible to build a model with up to 47x fewer and at least 4x smaller parameters, while still outperforming their original teacher model. This can significantly reduce deployment costs of these models and allows them to be effectively applied on low-power edge devices.

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

# A    APPENDIX

## A.1    COMPARISON OF POLICY DISTILLATION WITH DIFFERENT A2C TEACHERS

The policy distillation process requires the use of teacher networks, which are already fully trained, that transfer their knowledge into (smaller) student networks. In order to make our results more reproducible, we have chosen to make use of publicly available pre-trained checkpoints from the popular Stable Baselines3 (SB3) project (Raffin et al., 2019) for these teacher models. This project aims to provide reliable implementations for a variety of deep reinforcement learning algorithms that are representative for their respective capabilities. However, the published checkpoint for the A2C algorithm on the Atari Breakout environment at the time of writing does not reflect the performance that is reported in other state-of-the-art work. In our testing, this model obtains an average score of 248, while Schulman et al. (2017) report this to be 303, which was higher than obtained by their own PPO algorithm. When looking at the training logs associated with this checkpoint, a higher average score is in fact observed during training, but this is no longer the case for the checkpoint that was eventually published. We therefore retrained the A2C teacher model to make sure that our distillation procedure was not significantly impacted by this discrepancy, yielding a new average score of 342. We then compared our distillation results using this new teacher model to the results obtained using the checkpoint provided by SB3, resulting in figure 8. Note that this experiment was done by using $\lambda = 1$ in equation 2 to isolate the influence of the actual policy performance, leading to a deviation from the results show in section 6.2.

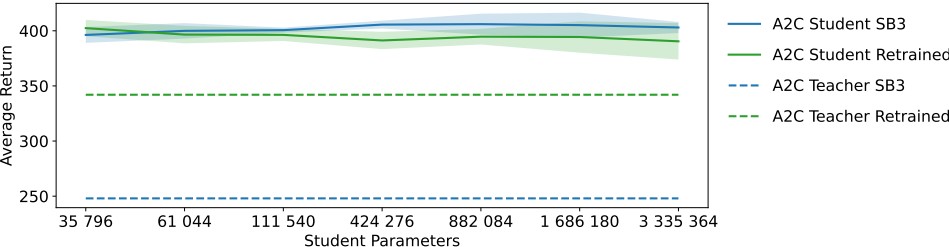

Figure 8: A comparison of the average score obtained on the Atari Breakout environment by students trained using a different A2C teacher models.

These results indicate that although the new A2C model does perform significantly better, there was no benefit to using this network as a teacher compared to the checkpoint provided by SB3. In fact, student performance was worse when using this teacher for all but the smallest network size. This is a clear testament of the regularization effect of policy distillation, by which the students using the teacher from SB3 are able to overcome some limitations of the teacher in its current state. The fact that the teacher which performs better itself is worse at transferring its knowledge to a student might seem contradictory, but as Stanton et al. (2021) have shown, KD does not typically work as commonly understood where the student simply learns to exactly match the teacher behaviour. There is a large discrepancy in the predictive distributions of teachers and their final students, even if the student has the same capacity as the teacher and therefore should be able to match it precisely. In this case, the output distributions provided by the SB3 teacher was a bit more informative for the students, even if the average return obtained for the corresponding actions and states was lower. For this reason and for the better reproducibility, we decided to use the pre-trained checkpoint provided by SB3 as an A2C teacher for all experiments throughout this paper, even though this teacher performs worse than what is reported in the state of the art.

## A.2    REAL-WORLD IMPROVEMENTS ON INFERENCE SPEED

In our experiments, we distilled three different teacher models into 7 student models of varying sizes to study the impact of the student capacity on the distillation performance. Our largest student model is 93x the size of the smallest one, but it remains to be seen whether this actually translates to a 9300% speed-up in real-world inference time. The relative improvement is heavily dependent on the class of device the model is running on, as seen in table 1. This table lists the average number of forward passes per second (FPS) obtained for each device and network type, all of which running

in 32-bit precision. These results are obtained by running a single observation through the network each time, as this more accurately reflects how the model will be used in deployment compared to running several observations at once, which can be parallelized more effectively. The PPO and A2C teacher models are combined in this table, as they use the same underlying network architecture. In these benchmarks, the critic head of the PPO and A2C models is also computed to give a completer picture, although this is usually not necessary for model deployment. If this head were omitted, they would instead share the same network architecture as the DQN and therefore perform equally in this benchmark.

Table 1: Average network forward passes per second for all used architectures and various popular low and high-power devices.

| Device | XXS | XS | S | M | L | XL | XXL | PPO / A2C | DQN |
|---|---|---|---|---|---|---|---|---|---|
| Raspberry Pi 3B | 135 | 128 | 116 | 100 | 72 | 46 | 25 | 39 | 41 |
| Jetson TX2 CPU | 79 | 78 | 73 | 67 | 52 | 45 | 32 | 42 | 45 |
| Jetson TX2 GPU | 815 | 808 | 815 | 816 | 762 | 760 | 736 | 280 | 568 |
| Intel NUC i5-4250U CPU | 1322 | 1287 | 1118 | 1071 | 706 | 535 | 381 | 388 | 487 |
| Nvidia GTX 1080 Ti GPU | 2560 | 2598 | 2610 | 2603 | 2599 | 2584 | 2541 | 1630 | 1811 |
| Tesla V100 GPU | 3620 | 3628 | 3638 | 3484 | 3466 | 3458 | 3625 | 2212 | 2477 |
| Ryzen 9 3900X CPU | 4535 | 4456 | 4327 | 3743 | 2649 | 2272 | 1442 | 1685 | 1934 |

Having more available CPU cores or access to a graphics card generally limits the relative speed-up that can be gained by applying policy compression, since there are fewer computations that can be performed in parallel for smaller networks. This can more clearly be seen in table 2, which shows the maximum relative speed-up for each device type. The only exception that sees a larger speed-up than some CPUs in our testing is the GPU that is part of the Jetson TX2, which was designed specifically for low-power operations.

Table 2: The maximal speed-up achieved on various devices by switching from the teacher to the smallest student model.

| Device | Maximal speed-up |
|---|---|
| Raspberry Pi 3B | 3.46x |
| Intel NUC i5-4250U CPU | 3.41x |
| Jetson TX2 GPU | 2.91x |
| Ryzen 9 3900X CPU | 2.69x |
| Jetson TX2 CPU | 1.88x |
| Tesla V100 GPU | 1.64x |
| Nvidia GTX 1080 Ti GPU | 1.60x |

Due to the lower potential to perform computations in parallel and fixed overheads from context switching, we actually observe far from the theoretical improvement on inference speed. An improvement of 3.46x from 39 to 135 FPS can still be significant however, as this is the difference between fluid motion and perceivable lag, especially when considering that in a real deployment there are additional processing steps required for each frame. The least powerful devices see the largest relative improvement, which are coincidentally exactly the target devices for which a lower inference time is essential for allowing these models to be deployed on the edge.

The exact network architecture is also impactful for the possible degree of parallelization, not only the number of parameters in the network. When running on a GPU device, our largest student network obtains a higher FPS than the teachers, although its number of parameters is higher. The structure of the network architecture therefore also needs to be carefully designed for operation on the target device before applying policy distillation.

### A.3    LOW-PRECISION DISTILLATION WITHOUT QPD

In this section, we demonstrate why the first two phases of our QPD algorithm 1 are essential to overcome the instability of training in low-precision, by performing an experiment where only the third phase is performed, using randomly initialized low-precision parameters. Aside from this, the same experimental setup is used as in section 6.3, with an A2C teacher and M sized student. Figure 9 shows the training loss of the third phase (QAT) for 5 independent students that were trained using only the last phase (non-QPD) and 5 students with the full QPD algorithm.

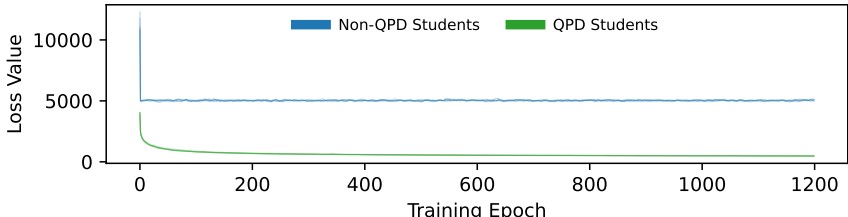

Figure 9: The QAT loss for 5 students trained using our full QPD algorithm compared to training a low-precision student directly using policy distillation.

The loss for non-QPD students drops steeply after the first training epoch, but the models fail to learn anything during subsequent epochs. The instability of training can also clearly be observed, as the loss is very noisy compared to the smoother descent in our QPD experiments. It also shows how our QPD loss starts at a point that is already lower than the non-QPD loss ever reaches, due to the parameters after applying PTQ already being much closer to their final values, even though the quantization introduced significant initial noise and a non-linear transformation was used.

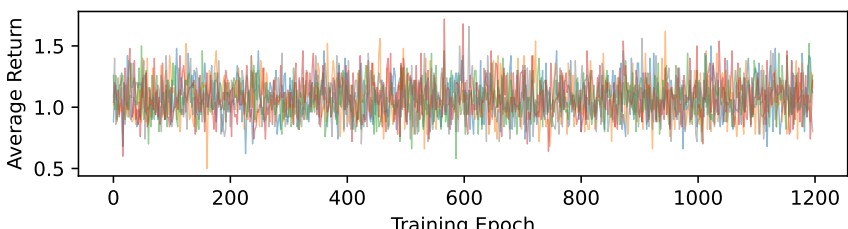

Figure 10: The mean return for 5 low-precision students trained directly using policy distillation.

Figure 10 depicts the mean return after each training epoch across 50 episodes obtained by these 5 students. This once again confirms that the students are not able to learn any useful behaviour when trained starting with randomly initialized low-precision parameters. The maximum mean return obtained by any of these students is 1.66, compared to 1.4 for a random agent and 412 for our QPD students. We can therefore conclude that a mechanism for transitioning more smoothly from the full-precision teacher parameters to low-precision student parameters is necessary to alleviate these optimization issues encountered when training in low-precision, for which QPD proved to be an excellent solution.

## A.4 NETWORK ARCHITECTURES

Tables 3 and 4 show the size of the models for the students and teachers used throughout our experiments. The architecture for our XS, S, M and XL students are the same as what was used in Policy Distillation (Rusu et al., 2016). On top of this, we added the architectures for our XXS, L and XXL students following a similar pattern to get a more complete picture of the influence these network sizes have on the distillation results.

Table 3: Sizes for the students used in our experiments.

| Network | Conv. 1 | Conv. 2 | Conv. 3 | Linear 1 | Compression Ratio | Parameters |
|---|---|---|---|---|---|---|
| Student XXS | 16 | 16 | 16 | 32 | 47.1x | 35 796 |
| Student XS | 16 | 16 | 16 | 64 | 27.6x | 61 044 |
| Student S | 16 | 16 | 16 | 128 | 15.1x | 111 540 |
| Student M | 16 | 32 | 32 | 256 | 4.0x | 424 276 |
| Student L | 32 | 64 | 64 | 256 | 1.9x | 882 084 |
| Student XL | 32 | 64 | 64 | 512 | 1x | 1 686 180 |
| Student XXL | 64 | 64 | 64 | 1024 | 0.5x | 3 335 364 |

The teacher architectures are the default policies provided by the Stable Baselines3 project (Raffin et al., 2019) for these algorithms. More specifically, the A2C and PPO network architectures are exactly the same, while the DQN architecture differs only by not having a critic head.

Table 4: The number of parameters in the teacher models used in our experiments.

| Teacher Model | A2C | PPO | DQN |
|---|---|---|---|
| Parameters | 1 686 693 | 1 686 693 | 1 686 180 |

