# OpenReview forum: "Quantization-aware Policy Distillation (QPD)"
_ICLR.cc/2023/Conference — Submitted to ICLR 2023_

### Official Review · Reviewer_zGhX · 2022-10-19

**Confidence:** 4
**Correctness:** 3
**Technical Novelty And Significance:** 2
**Empirical Novelty And Significance:** 3
**Recommendation:** 3

**Clarity, Quality, Novelty And Reproducibility:**

Since the proposed methods are rather simple, the clarity and reproducibility seem ok. However, the quality and novelty are my major concerns. From the view of model compression, the real contribution of this paper can be quite limited. The technique used in the paper has been common knowledge in network quantization and distillation (i.e., Section 4.1.1 and 4.2.2). Though the AUXILIARY CRITIC LOSS seems interesting, the experiments are too weak to support the contribution of this paper. I am not sure whether the proposed scheme could be generally effective in DRL.

**Strength And Weaknesses:**

# Strength:
- The results of "these combined methods can effectively compress a policy network down to 0.5% of its original size, without any loss in performance" seem encouraging. Though the authors mainly focus on the deployment, I personally think the proposed scheme may shed some light on the efficient training process of DRL, considering that the dual head architecture may serve as a self-distillation training pipeline to accelerate the convergence of both actor-network and critic-network. The idea of *AUXILIARY CRITIC LOSS* is interesting.
- The proposed method is quite simple and easy to follow, which may help to reproduce the results reported in this paper.
# Weaknesses:
- ### Major Comments:
  - As far as I can tell, the novelty of this paper can be limited given the current manuscript. The quantization and distillation methods have been proposed in previous model compression works [1], and are even widely discussed in the DRL community [2,3,4]. The only difference between QPD and pioneering approaches is the AUXILIARY CRITIC LOSS. However, the ablation study on the AUXILIARY CRITIC LOSS seems missing. It remains unclear whether the performance improvements come from the dual-head architecture or the CRITIC LOSS.
  - The authors elaborate on the benefits of "Distillation" $\rightarrow$ "PTQ" $\rightarrow$ "QAT". However, it seems that the authors use the same quantization strategy for “PTQ" and "QAT", which makes the Algorithm degrade into "Distillation" $\rightarrow$ "QAT". From my point of view, the overall training pipeline of Algorithm 1 is not new. It is common knowledge to equip "Distillation" with "Quantization" simultaneously or successively.
- ### Minor Issues:
  - According to Figure 7, it is obvious that network quantization results in negligible performance drops but the effect of distillation is hard to be found. Though Figures 4-7 have illustrated that "an auxiliary loss noticeably impacts the internal representation", more quantitative analysis on the improvements over strong baseline methods are encouraged, e.g., the real-runtime speed, memory footprint, model size, training time cost, and average return.
  - What if we further decrease the value of $\lambda$? Besides, it would be better to report the selection of $\tau$ in this paper.
  - I failed to find the descriptions of the experiment settings, such as the dataset, metrics, etc. Did I miss anything?
  - It seems that the authors conduct all experiments under the same circumstance. I expect more results reported on popular benchmarks with mainstream DRL methods.
  - "Our experiments show that these combined methods can effectively compress a policy network down to 0.5% of its original size, without any loss in performance" with a $200\times$ compression ratio seems inconsistent with Table 3.
  - It would be better to include a brief introduction to the "Atari Breakout environment" in Figure 1.

# Reference:
- [1] QKD: quantization-aware knowledge distillation. arXiv2019
- [2] Policy distillation. ICLR2016
- [3] Distillation strategies for proximal policy optimization. arXiv2019
- [4] Quantized reinforcement learning (QUARL). arXiv2019

**Summary Of The Paper:**

The authors propose to apply knowledge distillation and model quantization techniques to Deep Reinforcement Learning (DRL), which contributes to less memory footprint and space complexity during training. They show that the dual-head architecture leads to superior performance than the common actor smoothing and policy distillation. The experimental results reported in Figure 4 seem to further support the effectiveness of dual distillation.

**Summary Of The Review:**

The main track may not be the best venue for the current manuscript. The authors are encouraged to solve the weaknesses and re-submit this paper to the following conference.

---

> ### Author Response · Authors · 2022-11-18
> **Rebuttal Comment**
>
> The authors first thank the reviewer for their thorough review. We reply to your comments and issues below to hopefully provide some clarification.
>
> An ablation study of the auxiliary critic loss is included in the results and subsequent discussion of figure 4.
> When using a value of 1 for λ, our proposed actor-critic loss is reduced to only learning from the actor using the traditional policy distillation loss [1, 2], while still using a teacher with a dual-head architecture.
>
> Although existing work has indeed equipped distillation with quantization, as is also referred to in our related work [3], our QAT procedure does differ in a couple of key ways.
> In our algorithm, we keep using the full-precision parameters during the QAT phase and update these using a (full-precision) gradient computed based on the low-precision activations, instead of updating the low-precision parameters directly as is done in other work [3, 4]. Afterwards, we reapply quantization to propagate the gradient updates to the low-precision parameters.
> This layer of indirection with increased precision leads to a more stable optimization, while still taking the error in the activations imposed by quantization into account.
>
> We have included an analysis of the real-runtime speed and model size in the appendix and the storage memory footprint can be easily computed from this.
> The average return of our auxiliary critic loss is compared to existing work for distillation with actor-critic teachers [1] through λ = 1 in figure 4, and to a DQN baseline using the traditional policy distillation loss [2] in figure 6.
> The paper has now also been updated to include the selection of τ.
>
> The description of the experiment settings are explained in section 5: Experimental Setup.
> To summarize, we make use of a set of pre-trained models from the Stable-Baselines3 project [5] for our teachers.
> We evaluate our methods on the Atari Breakout environment that is part of the OpenAI Gym [6].
> The constructed dataset then consists of a replay buffer containing 540,000 transitions.
> After each training epoch, we replace the 10% oldest transitions with new teacher-environment interactions, by collecting observations and teacher outputs from new episodes until completion, while following the teacher's policy.
> As for metrics, we use the average return across 50 episodes after each training epoch, to a total of 600 epochs.
>
> We did indeed conduct all experiments using the same environment, but this was a deliberate choice that was also explained in section 5.
> A trade-off was made to use a single environment, but to include experiments on a wider range of DRL teacher algorithms, student sizes and two levels of precision.
> These experiments are more detailed than what is typically done in the state of the art, which we valued higher than performing the same limited set of configurations on a wider range of environments.
> We compare our results to the ones obtained by mainstream DRL algorithms in the form of the teacher results (Figure 1, 6 and 6), and to mainstream DRL compression and quantization methods by including a DQN distillation baseline [2], PPO actor-only distillation baseline [1] and by explicitly citing the results from Green et al. [1] and Krishnan et al. [4] in our discussion.
>
> Finally, table 3 only accounts for the number of parameters, but we also reduce the precision from 32-bit to 8-bit, resulting in a further 4x reduction in overall size.
>
> We again thank the reviewer for the many comments provided and for considering these additional clarifications.
>
>
> [1]: Green et al. Distillation strategies for proximal policy optimization. CoRR, abs/1901.08128, 2019
> [2]: Rusu et al. Policy distillation. ICLR 2016, Conference Track Proceedings, 2015.
> [3]: Kim et al. QKD: quantization-aware knowledge distillation. CoRR, abs/1911.12491, 2019.
> [4]: Krishnan et al. Quantized reinforcement learning (QUARL). CoRR, abs/1910.01055, 2019.
> [5]: Raffin et al. Stable baselines3. https://github.com/DLR-RM/stable-baselines3, 2019.
> [6]: Brockman at al. OpenAI Gym. CoRR, abs/1606.01540, 2016.

---

### Official Review · Reviewer_A7VX · 2022-10-25

**Confidence:** 3
**Clarity, Quality, Novelty And Reproducibility:** 1. Overall, it is a litte hard to fol…
**Correctness:** 3
**Technical Novelty And Significance:** 2
**Empirical Novelty And Significance:** 2
**Recommendation:** 5

**Details Of Ethics Concerns:**

N.A.

**Strength And Weaknesses:**

Weaknesses:
1. The distillation is decoupled with QAT process: they are conducted in a pipe-line fashion. Though author claimed that the submission is a smoother transition, a more valuable work would be solving the instability of `Quantization-aware Distillation'. Directly applying distillation-quantization makes the work less innovative.
2. Quantization is decoupled with reinforcement learning: the quantization phrase is more like an application of quantization (that works in other domains such as computer vision and neural language processing) to reinforcement learning. DoReFa and STE does not consider any properties that inherits in actor-critic models.

Strength:
1. The distillation method and analysis somehow enlight reinforcement learning understanding. Author may pay more attention to distillation part, instead of adding quantization if it is decoupled with reinforcement learning.

**Summary Of The Paper:**

This submission proposed to generated a compact and quantized version for actor-critic type reinforcement learning model. Basically, it first distilled a full-precision student model by introducing an auxiliary loss for critic model. Then the learnt model is served as initializer for Quantization-Aware Training (QAT).

**Summary Of The Review:**

Basically, author should 1) pay more attention to distillation, including analysis. 2) Get rid of quantization if it is just a plugin for the method. 3) make progress in integration of distillation and quantization in reinforcement learning scenario.

---

> ### Author Response · Authors · 2022-11-18
> **Rebuttal Comment**
>
> The authors thank you for reading through our paper and providing these insightful suggestions.
>
> We would first like to note that during the third phase of our algorithm, distillation is in fact still used to train the low-precision student as part of the QAT process, so there is no distillation → quantization pipeline as such.
> More accurately we could indeed state however that there is a high-precision → low-precision pipeline, with distillation being used at both stages.
> But even during the low-precision QAT stage, we still make use of both precision types to further increase the training stability, while still taking the error imposed by quantization into account for the weight updates.
> This is in contrast to other work where a similar high-precision → low-precision pipeline is employed [1][2], but where the second stage happens only with a single set of low-precision parameters.
>
> We do agree that solving the instability encountered when applying the distillation process directly in low-precision, thereby no longer requiring the first stage in that pipeline, would be of even greater value and something we plan to look into for future work.
>
> We thank you again for your review and for considering these additional comments.
>
> [1]: Kim et al. QKD: quantization-aware knowledge distillation. CoRR, abs/1911.12491, 2019.
> [2]: Krishnan et al. Quantized reinforcement learning (QUARL). CoRR, abs/1910.01055, 2019.

---

### Official Review · Reviewer_s1rF · 2022-10-26

**Confidence:** 4
**Correctness:** 2
**Technical Novelty And Significance:** 1
**Empirical Novelty And Significance:** 1
**Recommendation:** 3

**Clarity, Quality, Novelty And Reproducibility:**

The paper is poor in clarity, quality, and novelty.
I can not foresee the reproducibility of this paper.

**Strength And Weaknesses:**

Strengths:
1. A loss function is proposed to distill a dual-head network, which is suitable for actor-critic-based teacher networks.

Weaknesses:
Major
1. This paper claims they proposed a new Distillation loss and detail it in Section 4.1.
However, instead of using this proposed loss, the proposed QPD algorithm only uses the traditional distillation loss.
Do you have any reason for this?

2. The proposed loss simply combines two existing loss functions, and the novelty is limited.

3. The experiments can not evaluate the effectiveness of the proposed methodology.

4. The writing should be improved since some sentences and tables are confusing.

**Summary Of The Paper:**

The paper proposed a new loss function that combines KL loss and Huber loss and can be used in actor-critic-based teacher networks.
They designed a 3-phases algorithm to distill knowledge from a full-precision teacher network to a low-bit student network.
They discussed the influence of the choice of teacher algorithm.

**Summary Of The Review:**

This paper proposed a new loss function that can be used in the dual-head network.
But the writing is poor, and the novelty is limited.
So I recommend rejecting this paper.

---

> ### Author Response · Authors · 2022-11-18
> **Rebuttal Comment**
>
> The authors thank you for thoroughly reviewing our paper and suggesting areas where it could be improved.
>
> First, to answer the question posed in the review:
> Our proposed QPD algorithm does in fact use the distillation loss detailed in section 4.1.2 for actor-critic networks, as is also referred to in our algorithm overview (4.2.1).
> The reason we referenced the traditional distillation loss in Algorithm 1 is because we also apply this for a DQN teacher, which is not compatible with the proposed actor-critic loss.
> This made the formal description of our algorithm more generally applicable, at the cost of some potential confusion about our precise methodology.
> Thanks to your review, we have updated Algorithm 1 in our paper to reference the proposed actor-critic loss instead of the traditional distillation loss to hopefully make this more clear.
>
> Although the implementation of combining the existing loss functions is indeed straight-forward, the concept of using an auxiliary loss to enhance the internal representation of a model has, to the best of our knowledge, not yet been studied in the context of single-task knowledge distillation.
> We would argue that our contribution not only exists in defining this loss and being the first to produce results that exceed the traditional distillation loss baseline [1] [2], but also in demonstrating what tangible effect this has on the learned internal representation.
> Our closer investigations into the effects the temperature parameter has on the dark knowledge that is actually distilled also results in a difference between our approach and those reported in existing work [1].
>
> For the possibility of future improvement, we would also like to ask some clarification on how our experiments are not able to evaluate the effectiveness of the proposed methodology?
> We first compared our proposed actor-critic distillation loss to a baseline where only the actor is distilled, corresponding to the setup proposed by Green et al. [1], to evaluate the effectiveness of learning from the auxiliary state-value function.
> Then, a comparison is made to a DQN baseline to evaluate the effectiveness of distilling a policy-gradient based teacher compared to the traditional distillation loss [2].
>
> We also thank the reviewer in advance for reading through these additional comments.
>
> [1]: Green et al. Distillation strategies for proximal policy optimization. CoRR, abs/1901.08128, 2019
> [2]: Rusu et al. Policy distillation. ICLR 2016, Conference Track Proceedings, 2015.

---

### Decision · Program_Chairs · 2023-01-20

**Decision:**

Reject

**Justification For Why Not Higher Score:**

N/A

**Justification For Why Not Lower Score:**

N/A

**Metareview: Summary, Strengths And Weaknesses:**

This paper received controversial appreciations: it implements an interesting underlying idea, especially in the context of distillation and reinforcement learning.
However, there are common criticisms regarding the weak novelty, especially in relation to distillation (which is deemed standard), limited technical contributions, insufficient experimental analysis, including ablation study, and also about the clarity of the presentation.
Authors provide a rebuttal, but reviewers are still considering the paper not ready for publication, all giving ratings below threshold.
For these reasons, this work cannot be accepted for publication to ICLR 23.


**Summary Of Ac-Reviewer Meeting:**

N/A